# Tackling HLA Deficiencies Head on with Oncolytic Viruses

**DOI:** 10.3390/cancers13040719

**Published:** 2021-02-10

**Authors:** Kerry Fisher, Ahmet Hazini, Leonard W. Seymour

**Affiliations:** Department of Oncology, University of Oxford, Oxford OX3 7DQ, UK; Ahmet.Hazini@oncology.ox.ac.uk

**Keywords:** oncolytic virus, class I HLA, immunosurveillance, immunotherapy

## Abstract

**Simple Summary:**

Oncolytic viruses show great promise as anticancer agents by simultaneously lysing cancer cells while stimulating innate and adaptive immune responses. However, the extent to which the adaptive immune system contributes to overall efficacy alongside oncolytic viruses is, in part, dependent on the compliance of cancer cells to present antigens correctly. Dysregulation of any part of the antigen presentation machinery provides a strong selection pressure for immune escape. In this review, we consider the key immunological factors that might be measured to allow for the optimum deployment of oncolytic viruses for effective cancer therapy.

**Abstract:**

Dysregulation of HLA (human leukocyte antigen) function is increasingly recognized as a common escape mechanism for cancers subject to the pressures exerted by immunosurveillance or immunotherapeutic interventions. Oncolytic viruses have the potential to counter this resistance by upregulating HLA expression or encouraging an HLA-independent immunological responses. However, to achieve the best therapeutic outcomes, a prospective understanding of the HLA phenotype of cancer patients is required to match them to the characteristics of different oncolytic strategies. Here, we consider the spectrum of immune competence observed in clinical disease and discuss how it can be best addressed using this novel and powerful treatment approach.

## 1. Introduction

Cancer is a disease that constantly adapts to remain hidden beneath the immune surveillance radar. Cells acquiring mutations or changes that are noticed by the immune system can be pruned away by T-cell immunosurveillance, effectively providing a threshold governing which changes are sufficiently stealthy to persist. However, under constant selective pressure, tumors often develop a capacity for subversion, for example by the upregulation of TGF-β or immune checkpoints, effectively raising the immunosurveillance threshold which allows a greater range of genetic and epigenetic changes (tumor associated antigens (TAAs)) to persist without detection [1].

We can counter this immune subversion strategy in a significant minority of patients using immune checkpoint inhibitors that can reveal cancer cells, once again, as legitimate targets for destruction by the immune system. However, the majority of patients do not respond to checkpoint inhibition, and those that do can eventually develop resistance [2]. This suggests there are multiple mechanisms of immune evasion and a constantly evolving battleground between tumor and immune cells. In many cases, cancer cells may eventually acquire the capacity to turn off the immune surveillance radar completely by usurping the class I HLA system to protect themselves from detection (Figure 1).

Without class I HLA presentation, T-cells cannot recognize and kill target cells, even if all other aspects of the immune system are fully functional [3,4]. The majority of immunotherapy approaches currently under development—including cancer vaccines, checkpoint blockade, adoptive T-cells and STING (stimulator of interferon genes) agonists—are rendered completely obsolete without functional class I HLA presentation by cancer cells. Any aspirations for encouraging epitope spreading [5] or inciting a systemic abscopal effect are likely to become redundant if key elements of the HLA pathway are sufficiently compromised.

## 2. The Murky and Complex World of HLA Deregulation

There are a myriad of ways that class I HLA expression can be downregulated, corrupted or made dysfunctional [6,7,8,9,10]. A frequently observed event in cancer is loss of heterozygosity (LOH) in the HLA locus, meaning that one allele is lost, effectively providing cancer cells with a wider operational envelope to accumulate TAAs without detection. For example, LOH in the HLA locus of lung cancer can cause the loss of HLA-C*08:02, meaning that driver mutations like *K-RAS* (Kirsten rat sarcoma) G12D are no longer presented and G12D-specific T-cells are no longer effective [11]. The frequency of this change in cancer in early-stage lung cancer is reported to be 40% [12], emphasizing the importance of this escape mechanism to mask immunogenic TAA.

Non-silent genetic mutations in class I HLA genes (including *HLA-A*, *-B*, *-C* and *beta-microglobulin* (*β2M*)) are reported at low frequencies in many cancers, typically <5% [13], but in microsatellite unstable colorectal cancer (MSI-H CRC), where there is a strong immune pressure, this can be as high as 30% [14]. However, focusing on mutations directly in HLA genes alone will only reveal a small fraction of the problem because any gene involved in the antigen presentation pathway (e.g., proteasomal processing, peptide loading in the ER or regulatory genes such as *NLRC5*) can result in deficient class I HLA function [14]. The cumulative impact of all these molecular changes, together with heterogeneity between clones of tumor cells, adds up to a complex picture. Given the lack of any established biomarkers for HLA functionality in patients, the importance of genetic mutations is likely not fully appreciated.

Defects arising from genetic mutations or allelic loss are often referred to as irreversible or hard defects [9]. In contrast, soft defects are epigenetic in nature and are potentially reversible through pharmacological intervention. Causes of soft HLA defects are varied and include deficiencies in interferon pathways [15] or the hypermethylation of key HLA regulatory elements [16,17]. They are much harder to study and quantify than genetic changes and may reflect responses to applied therapeutic immune pressure or be related to other features of cancer progression and the tumor microenvironment including TGF-β signaling [18], ER stress [19] and hypoxia [20]. Upregulation of class I HLA can be achieved, in principle, through interferon signaling due to an interferon response element (ISRE) in the promoter region of all the classical HLA molecules [15,21,22,23], although this approach may not be effective in all cancers due to frequent deregulation of interferon pathways [24,25,26].

The loss of type I HLA expression in cancer cells is often seen as an invitation for elimination by NK cells. However, in the complex evolutionary environment of the tumor, NK cells provide just one more challenge to work around. In consequence, the simple absence of class I HLA is not likely to be a common occurrence; tumor cells that lose their classical class I function alongside the upregulation of the non-classical HLA molecules HLA-G and/or E, which do not provide classical antigen presentation but can inhibit the activation of NK cells [27,28], are likely to be more successful. Tumor heterogeneity may apply to HLA function in the same way that it applies to mutational load, meaning that different tumor cells and their progeny may acquire different HLA deregulation strategies, particularly in the face of immunological therapies. In other words, we should not think in terms of binary HLA loss, but of a constant bio-selection to maintain a balance of HLA expression appropriate for the continued existence of a population of cancer cells. For further details on HLA loss and the underlying mechanisms, please refer to an earlier review in this journal [6].

## 3. Treating HLA-Competent Tumors with Oncolytic Viruses or Immunotherapy

Cancers with functional HLA should be relatively amenable to a wide range of immunotherapies, although this can probably occur only if they are infiltrated with functional CD8^+^ T-cells (Figure 2, category A). In these patients, a single adjustment to the immune system through either checkpoint inhibition or oncolysis could arguably be sufficient to reach a tipping point that enables the immune system to mount an effective response. The first approved oncolytic viruses for human use, Imlygic, lends support to this concept with significant numbers of responses that appear to be immunological in nature [29]. For tumors showing an abscopal response (>50% shrinkage of non-injected lesions), comprising of about 34% of superficial lesions and 15% of visceral lesions, it would be reasonable to hypothesize that the HLA system remains at least partially functional. The addition of checkpoint inhibitors with Imlygic has further increased the response rate [30,31], perhaps reflecting their independent mechanisms of action that may synergize to allow HLA-mediated cytotoxicity. Whether this is sufficient to extend patient survival compared to either treatment alone is currently being evaluated in a phase 3 clinical trial (MASTERKEY-256/KEYNOTE-034). Meanwhile several other oncolytic agents are being explored in combination with checkpoint inhibitors, including Cavatak, Reolysin, MG1-MAGEA3, ONCOS-102, DNX-2401, HF-10, Pexa-Vec and Enadenotucirev [32]. For all these trials, it would be very helpful to prospectively correlate patient HLA expression with clinical observations, although we are not aware that any are planning to do so.

Where functional HLA presentation on cancer cells is confirmed, it is compelling to pursue oncolytic viruses as in situ personalized vaccines, releasing and exposing TAAs upon lysis [33,34,35]. This effect is amplified by triggering immunogenic cell death (ICD) pathways that induce wide-ranging immunological consequences, including the maturation of dendritic cells and activation of T-cells [36,37,38,39,40]. On top of the inherent ability of viruses to induce ICD, arming them with transgenes can further stimulate different arms of the immune system through the careful choice of cytokines or checkpoint inhibitors, among myriad other possibilities [41].

## 4. Boosting Lymphocyte Uptake into HLA Competent Tumors

An important and unusual benefit of oncolytic viruses is the potential to attract CD8^+^ cells that may otherwise be excluded from tumors, a phenomenon particularly apparent in carcinomas [42,43,44]. Promoting lymphocyte engraftment into tumors could be essential to allow for the initiation of an effective immune response, although it can only be useful if type I HLA expression is functional (Figure 2, categories A–D). Lymphocytes are attracted towards a chemokine concentration gradient and replicating lytic viruses, established within the tumor, are well placed to become a homing beacon for immune cells [31,45,46,47]. It is also possible that new CD8^+^ cells entering the tumor may have greater activation potential than the endogenous T-cells that may have become anergic following exposure to the tumor microenvironment (TME) for an extended period. This phenomenon should place oncolytic viruses as ideal partners for other immunotherapy approaches, since mobilizing immune cells in cancer patients by other approaches is far more challenging. For example, administering immune stimulants like STING agonists, chemokines, or interferons directly into the blood stream is likely to provide only a relatively short period of activation at the expense of systemic side effects, without any locoregional information to guide immune cells towards the tumor [48,49].

To further augment the capacity of oncolytic viruses to promote immune engraftment, it is possible to arm them to express chemokines within the tumor [50,51,52]. In this context, particularly desirable oncolytic viruses might be those that that persist locally and express chemokines for extended periods of time. However, attracting cytotoxic T-cells into the locality of a replicating oncolytic virus brings with it the capacity for rapid recognition and elimination of the virus itself, and a consequent premature end to the therapy [53]. The subtle interplay between viruses and the HLA system has been evolving for millennia, and the implications for oncolytic virus design are considered in the next section.

## 5. The Consequences of Removing HLA-Manipulating Viral Proteins from Oncolytic Viruses

Many viruses contain proteins, such as E3 19k in adenovirus or ICP47 in HSV, that are capable of corrupting antigen presentation inside infected cells using strategies as diverse as blocking the TAP transporter or pulling HLA molecules away from the cell surface [54]. These elements have been removed from many oncolytic virus candidates with the intention of making oncolysis more immunogenic, and to improve safety by accelerating viral clearance in normal tissues [55,56]. However, in hot tumors, broadcasting a cancer cell as virus-infected in this way may lead to premature virus elimination by the immune system, necessitating repeated virus delivery in the face of increasing levels of neutralizing antibodies or restricting treatment options to direct intratumoral injection. Conversely, in cold tumors that lack CD8^+^ cells, this may be less of an issue, at least initially, with virus infected cells only becoming targets after immune cell infiltration is restored.

Removing HLA-inactivating genes from oncolytic viruses might also be counterproductive to creating a durable anticancer immune response. One of the benefits of oncolytic viruses is the capacity to cause immunogenic cancer cell death, shedding adenosine triphosphate, heat shock proteins and other immune-provoking signals alongside cancer antigens into the interstitial space [37,40,57]. This could create a proinflammatory environment, potentially leading to an anticancer immune response. In contrast, allowing the presentation of virus antigens on HLA will likely lead to efficient T-cell-mediated killing of infected cancer cells by caspase-mediated apoptosis, regarded as a less inflammatory death mechanism [58]. Arguably, this could be less effective at priming new anticancer immune responses than virus mediated lysis with the simultaneous presentation of pathogen or damage associated molecular patterns (PAMPs and DAMPs).

Cancer cells infected with an oncolytic virus are likely to be destined for eventual elimination, either through lysis or following the exhaustion of metabolites. Rather than killing them rapidly via T-cell cytotoxicity, it may be preferable to allow virus infected cells to persist as a factory, modifying the tumor microenvironment and releasing TAAs, while T-cells kill any residual cells not infected by the virus [59]. Accordingly, cancer selective viruses that can avoid T-cell killing may have the advantage, especially when relying on the adaptive immune response as the major driver for efficacy.

## 6. Treating Cancers with Reversible HLA Defects

Where a cancer cell lacks functional class I HLA, little benefit can accrue from checkpoint inhibition or attracting lymphocytes into the tumor microenvironment. Efforts to express or expose TAA are likely to be fruitless until HLA function is restored. Turning HLA defective (soft) cancers (Figure 2, categories E–H) into HLA competent tumors (Figure 2, categories A–D) is therefore an attractive proposition and can exploit several strengths of the oncolytic approach, particularly where those soft mutations are mediated through deficient interferon pathways. In particular, viruses have an intrinsic capability to induce interferons during infection and lysis by triggering pathogen recognition receptors such as cGAS (cyclic GMP-AMP synthase) or RIG-I (retinoic acid-inducible gene I) in cancer cells or adjacent stromal cells [60,61]. Viruses can also be armed to express additional levels of interferons selectively in the tumor microenvironment [62,63,64].

Although soft HLA defects are not understood in detail in patients, they feature strongly in laboratory models. For example, B16 cancer cells are a classic example of soft class I HLA defects, with low levels of HLA that can be fully restored by treatment with gamma interferon [65]. It is intriguing to ponder whether it is interferon-induced class I HLA upregulation rather than anything else that leads to efficacy with oncolytic viruses in this model, given that persistent lysis would not be required in the presence of an anticancer immune response [66].

This potential for the locoregional upregulation of class I HLA expression in tumors, using interferon either encoded within the virus or produced naturally following virus infection, should synergize with the ability of oncolytic viruses to attract and activate T-cells. In turn, this could condition the tumor microenvironment to support additional immune therapies like checkpoint inhibitors that have little chance otherwise of having an impact. Accordingly, oncolytic viruses could be the key to enabling the effective immunotherapy of cancers with reversible HLA defects (E to H).

While upregulating classical HLA-A, -B and -C in tumors with soft mutations may be highly desirable to engender an adaptive immune effect, care must be taken not to upregulate non-classical HLA-G/E, which have a variety of mechanisms to inhibit classical antigen presentation. Variations in the promoter regions of HLA-A/B/C vs. HLA-G/E may allow pharmacological approaches to achieve this [67]. Careful analysis of the regulation of classical and non-classical HLA molecules, together with the versatility of oncolytic viruses, could permit the subtle manipulation of HLA expression to achieve the greatest therapeutic impact.

## 7. Implications of Interferon Competency in Cancer Cells with HLA Reversible Cancers

Cancer cells with defects in interferon pathways or NF-κB signaling pathways may not be amenable to the correction of HLA function via the expression of interferons (Figure 2, categories G,H). This is crucially important for some oncolytic approaches that are designed to exploit cancer cells that have deregulated interferon pathways [68]. At face value, these oncolytic agents would perhaps not be the first choice for patients with interferon-reversible soft HLA defects because, by definition, these tumors are likely to be interferon responsive and therefore the activity of these viruses would be limited.

However, the reality is more nuanced, because defects in interferon signaling can be at different stages of the pathway, with some cancer cells not able to express interferons and others not able to respond to them. In a stromal-rich tumor, fibroblasts and macrophages are likely to have the full capacity to detect oncolytic viruses and express interferons even if the cancer cells are defective [69]. Cells that can be stimulated by external interferon (with intact JAK/STAT signaling) would often have functional interferon regulatory factors (IRFs), and therefore be able to trigger ISRE to upregulate classical class I HLA expression. Conversely, cells with deficient JAK/STAT pathways could likely still upregulate HLA, but only if their own virus-sensing pathways remain intact. Consequently, a major subset of cancer cells with functional JAK/STAT pathways but defects in virus sensing and interferon expression, could likely upregulate HLA in response to interferons generated within the TME by an oncolytic virus interacting with stromal cells. However, this may be to the detriment of further oncolytic virus spread and lysis. Accordingly, matching the patient population to the oncolytic strategy is very important.

Some DNA viruses have sophisticated mechanisms to overcome interferon responses, for example the E1A protein and the VA RNAs in adenovirus, B8R and B18R/B19R in vaccinia, and ICP34.5, US11 and others in HSV-1 [70]. Adenoviruses, for example, can continue replicating despite triggering interferon and STING pathways, allowing both immune stimulation and lysis to happen concurrently [71]. When DNA viruses are genetically attenuated to render them interferon sensitive, the innate immune response works to limit viral replication [72]. Consequently, patients that are considered to have interferon-inducible HLA may be preferentially matched with viruses that can operate in a wider variety of interferon competent environments. This may be particularly true in heterogeneous tumors where there is likely to be a variety of different interferon defects in different populations of cancer cells, and also in those cancers with a high stromal cell content. In contrast, for viruses that are unavoidably dependent on dysfunctional interferon, it may be possible to achieve upregulation of HLA via interferon-independent pharmacological means such as HDAC inhibition [73].

## 8. Oncolytic Viruses for the Treatment of Irreversible Hard Defects

In cancers where there are genetic defects in the HLA pathway, attempts to mount a comprehensive adaptive immune response are likely to be futile. In these cancers (Figure 2, categories I–L), decisions over how best to invoke ICD or stimulate antigen presenting cells becomes straightforward because none are likely to be successful. For these cancers, the full spectrum of oncolytic mechanisms and arming strategies will need to focus on achieving cytotoxicity independent of HLA expression, and this embodies the essence of the whole concept of direct virolysis. Prior to discussing the range of different HLA-independent cytotoxic strategies available to oncolytic viruses, it is worth challenging the very notion that hard HLA defects are definitively irreversible. Oncolytic viruses have the capacity, in principle, to normalize HLA expression in any cell they infect.

### 8.1. Gene Supplementation Therapies

Wild type copies of each of the components of the antigen presentation pathway can in theory be encoded within oncolytic viruses and expressed locally within tumor cells to restore pathway function. Although this approach could be adopted for any pathway component, perhaps the most widely studied is β2M [74], where gene replacement with adenovirus has been shown to restore cell surface expression of classical class I HLA molecules, implying that β2M function has been restored [75]. Importantly, the restoration of β2M was also shown to lead to peptide-specific immune recognition by cognate antigen-specific T-cells [76].

Unfortunately, the therapeutic benefits of direct class I HLA gene replacement therapy will likely be very limited, because current understanding suggests that the expressed transgene product would be restricted only to cells that are actively infected with the oncolytic virus. This raises two immediate concerns—first, that functional antigen presentation would not be restored more broadly within the tumor, but only in cells that would be expected to be killed by direct oncolysis, and secondly that the restored HLA function in those infected cells might begin to present viral epitopes rather than TAAs, giving rise to an augmented antiviral T-cell response rather than stimulating an anticancer response. This might accelerate the immune-mediated clearance of the virus rather than empowering a cancer vaccination effect. Hence, the concept of replacing mutated components of the antigen presentation pathway appears flawed unless it can be somehow delivered more broadly within the tumor and not restricted just to virus-infected cells. One possible approach that might be worth exploring is the use of exosomes. Although still in its infancy, the potential for programming viruses to manipulate exosomes to distribute functional HLA molecules amongst cancer cells may be an effective way of restoring the presentation of TAAs to allow renewed immunosurveillance.

### 8.2. Turning Lymphocytes into HLA-Independent Killers with Virus-Deployed Bispecific T-Cell Engagers

Bispecific T-cell engagers (BiTEs) crosslink endogenous CD3 on T-cells to surface targets on cancer cells, creating an activating pseudosynapse through clustering and leading to rapid and efficient target-specific cytotoxicity. The T-cells then detach from the target cell and can bind to a new target cell, earning them the epithet serial killer T-cells. This powerful approach is reminiscent of converting endogenous T-cells into antigen-specific chimaeric antigen receptor (CAR) T cell-like cells in situ, recognising any chosen surface antigen.

BiTEs are difficult to deploy through conventional intravenous delivery because of their short plasma half-life, and fine tuning of affinity of the CD3 binding arm is crucial to prevent sequestration to T-cells outside the tumor. When it comes to the choice of the antigen-binding arm for a systemic therapy, there is a trade-off between using broadly expressed antigens that could mediate on-target off-disease side effects, or highly cancer-selective antigens that may not be universally expressed throughout the tumor. Expressing BiTES locally and exclusively in the tumor microenvironment from an oncolytic virus may avoid these complications, allowing for a far broader range of target and effector arms to be considered. A constantly renewed supply of BiTEs might be expected to synergize with newly arriving T-cells encouraged to the tumor through chemokines or oncolysis.

Several groups have now demonstrated the successful delivery of BiTEs using a range of oncolytic viruses. The earliest approaches used oncolytic vaccinia and adenovirus to express BiTEs targeting receptors on the cancer cell surface, allowing cancer cells to be targeted simultaneously by two distinct cytotoxic entities—the oncolytic virus and the BiTE-targeted T-cell [77,78,79]. It would be difficult to use any other strategy to target these surface markers with the anatomical selectivity provided by oncolytic viruses. However, despite the potency and selectivity of the approach, it was soon considered suboptimal because the rapid cytotoxicity can decrease the pool of therapeutic viruses by eradicating the tumor cells that produce them. Accordingly, more recent innovations have explored combining the direct cytotoxicity of oncolytic viruses against cancer cells with BiTEs that retarget endogenous T-cells to attack other cellular components of the tumor microenvironment. To date, these have included tumor-associated macrophages via targeting to folate receptor b or CD206 [80], or tumor-associated fibroblasts via fibroblast activation protein (FAP) [81,82]. This latter approach forms the basis of an ongoing clinical trial which uses an oncolytic adenovirus to express a FAP BiTE alongside alpha interferon (which should increase immune stimulation and may restore HLA function), together with two chemokines intended to recruit T-cells into the tumor (Clinical Trials identifier NCT04053283). Finally, harnessing endogenous T-cells in this way should provide greater therapeutic potential for heterogeneous solid tumors than exogenously applied CAR T-cells, since every T-cell within the tumor may be redeployed against the chosen target antigen, whereas CAR T-cells will always be a subset of those engrafting into solid tumors.

## 9. Orchestrating Anti-Cancer Innate Immune Cells in the Absence of Functional HLA

A great strength of oncolytic viruses is that they are fully customizable drugs that can be fine-tuned to accomplish the tasks at hand. Where adaptive immunity is unlikely to be harnessed, arming elements targeting cancer vaccine approaches can be swapped out for those targeting the innate immune system. NK cells have been directly implicated in enhancing the efficacy of oncolytic viruses in experimental models [83]. Arming oncolytic viruses with IL-15 [84] is likely to augment NK activation, while strategies based on CCL5 chemotaxis [85] have been used to attract them in to tumors. Expression of SIRPa-Fc antagonists from oncolytic viruses can block the CD47 “don’t eat me” signals on cancer cells and facilitate macrophages to attack them directly [86]. Building on the BiTE principle, it is now possible to develop bispecific killer cell engagers (BiKEs) that crosslink cancer cell surface antigens with CD16 on NK cells, or better still, trispecific TriKEs that included further functionality through integrating IL-15 [87,88]. Equivalent approaches using BiMEs (bispecific macrophage engagers) to activate macrophages have also been reported [89]. Like all arming approaches to activate the adaptive immune system, strategies for exploiting innate cells will benefit from selective expression in the tumor microenvironment provided by oncolytic viruses. Careful design of the innate immune cell stimulation is required in order to avoid the premature and unwanted elimination of virus-infected cells.

## 10. Strategies to Maximize Cell Killing by Oncolytic Viruses

Oncolysis is an HLA independent killing mechanism that should be exploited in cancers with severe immunological defects. Several hallmarks of cancer, including immune deregulation, appear to provide an advantageous niche for virus activity [90]. These also include the intrinsic resistance of cancer cells to apoptosis and their reprogrammed energy metabolism that provides biosynthetic intermediates to support macromolecular synthesis. When this is combined with hard HLA defects and little to no prospect of a functional cytotoxic T lymphocyte response, a virus might be expected to remain on station for as long as there are substrate cancer cells to infect. In xenografts, efficacy through oncolysis can be demonstrated without help or hindrance from T-cells [91,92,93].

The combination of virotherapy with chemotherapy can also lead to enhanced virus spread and cell lysis. Synergy with a wide range of chemotherapeutic agents, including platinum drugs, taxanes and topoisomerase inhibitors, among others, have been reported and reviewed in detail elsewhere [94]. Arguably, the most convincing anticancer efficacy in solid carcinomas with oncolytic viruses was in combination with chemotherapy [95]. In this study, patients refractory to several lines of previous treatment were injected intratumorally with ONYX-015 concurrently with intravenous cisplatin and 5FU. Only one of the tumor nodules (the largest) was injected, with uninfected tumors acting as internal controls. In 19 evaluable patients, eight had a complete response in the injected tumor and the remainder had a partial response with signs of extensive lysis. That this effect was mediated by direct oncolysis without adaptive immunity was evidenced by the lack of any effect on non-injected tumors.

Arming viruses to treat HLA compromised tumors can be fully focused on promoting intratumoral virus spread and direct oncolysis. This is made easier because virus gene products that evade premature CTL killing like E3 19k in adenovirus or ICP47 in HSV-1 become redundant in this setting and removing them makes space for additional transgenes. Matrix degrading enzymes or cell fusion peptides have both demonstrated enhanced spread of oncolytic viruses in vitro and in vivo. Enzyme prodrug therapy using thymidine kinase (TK) or cytosine deaminase (CD) [96] is an alternative approach that combines many of the benefits of targeted chemotherapy with the anatomical selectivity of a virus. An innocuous prodrug given systemically that is activated into a cytotoxic entity only by the enzyme within the tumor restricts cytotoxicity to the environs of the infected tumor cells and spares normal tissues, including the bone marrow. In the case of TK, the prodrug is usually ganciclovir, which is metabolized to yield a potent DNA polymerase inhibitor, whereas CD metabolism 5-fluorocytosine to become 5-fluorouracil (5FU), which is a widely used inhibitor of thymidylate synthase. In particular, 5FU is an attractive therapeutic for this approach because it diffuses widely and will mediate a considerable bystander toxicity against cells that are not directly virus infected [97]. Future strategies to enhance virus spread or cytotoxicity could involve transgene products, including proteins or nucleic acids packaged into exosomes and other membrane vesicles that are shed from the infected cell, and thereby allow their transfer into the cytosol of uninfected cancer cells nearby [98].

## 11. Conclusions

Progress in cancer immunotherapy over the past two decades has been extraordinary, transforming the lives of many patients that were previously incurable. Intense ongoing research is focused on the limits of our ability to re-educate the immune system and turn it against tumor antigens. Success in the clinic, together with more sophisticated animal models that allow for the demonstration of novel immunotherapies, has attracted much of the global capacity for oncology research and development.

Yet despite this, the majority of patients do not respond to immunotherapy. This failure may continue unless we tackle the underlying features of immune evasion head on. HLA deficiency is arguably the primary culprit because deregulation in any of the genes or pathways involved in antigen processing or presentation will thwart any attempts at perpetuating the clearance of cancer cells via the adaptive immune system. Current laboratory tests for class I HLA expression do not take into account the presence of non-classical HLA molecules (notably HLA-E and -G), and hence cannot reliably assess HLA function. It follows that a deeper understanding of a patient’s capacity for adaptive immune responses, including HLA functionality, will be essential to allow as much clinical impact as possible.

Oncolytic viruses are supremely versatile anticancer agents that have the capacity to address some of the greatest therapeutic challenges and unmet needs for patients. This includes exploiting class I HLA where it is functional, or inducing it where it is not, provided we can accurately identify patients who would benefit in those scenarios. Despite the inherent ability of viruses to induce interferons and potentially restore or upregulate HLA, surprisingly no primary oncolytic paper has focused on this area. In many cases, this might be because cancers with functional interferon pathways have been avoided for pre-clinical research using those oncolytic agents that are interferon sensitive. On a broader point, a greater awareness of the HLA status of animal tumour models could be very useful to help interpret preclinical therapeutic activity and might contribute towards stratification of patients suitable for different types of treatment.

Perhaps the most powerful approach of all is to use the characteristics of oncolytic viruses to invoke HLA independent immunotherapy or mediate direct cytotoxicity alone, or perhaps in combination with chemo/radiotherapy. Although this is particularly relevant to cancers with severe immune deregulation, such a strategy could be applied more broadly without having to be overly concerned with HLA status.

It follows that oncolytic viruses have great potential to contribute to meaningful therapies for patients with any status of immune function. However, which oncolytic approach will be most successful depends on both the class I HLA status and the interferon competence of the tumor, including whether there is clonal heterogeneity between metastases or even within individual tumors. Knowledge of HLA functionality and interferon status in individual patients is essential to guide the choice of optimal treatment strategy, but that information is currently very hard to obtain. A laboratory test for HLA function that could be performed on biopsies would revolutionize our ability to deploy oncolytic and other immune stimulatory strategies effectively.

## Figures and Tables

**Figure 1 cancers-13-00719-f001:**
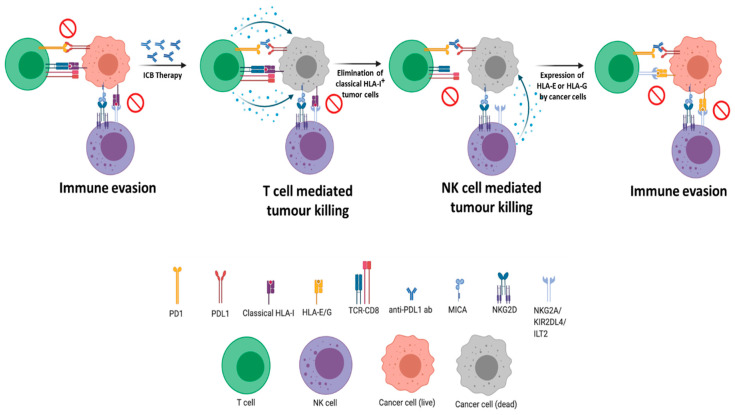
Overview of immune evasion strategies employed by tumor cells. T-cell mediated killing requires functional class I HLA and can be inhibited by tumor expression of immune checkpoints such as PDL1 (programmed death ligand 1). Immune activity in this situation can be restored using checkpoint inhibitor antibodies such as anti-PDL1. Similarly, NK (natural killer) cell-mediated cytotoxicity requires engagement via an NK cell ligand such as MIC-A (major histocompatibility complex class I polypeptide—related sequence A), coupled with the absence of class I HLA. Expression of the non-classical HLA haplotypes HLA-E or HLA-G (which are not generally recognized by the TCR (T cell receptor)) provide a simple mechanism for tumor cells to evade killing by both T-cells and NK cells. Additional abbreviations: ICB (immune checkpoint blockade), NKG2D (natural killer cell receptor G2 type D), KIR2DL4 (Killer cell immunoglobulin-like receptor 2DL4) and ILT2 (immunoglobulin-like transcript 2).

**Figure 2 cancers-13-00719-f002:**
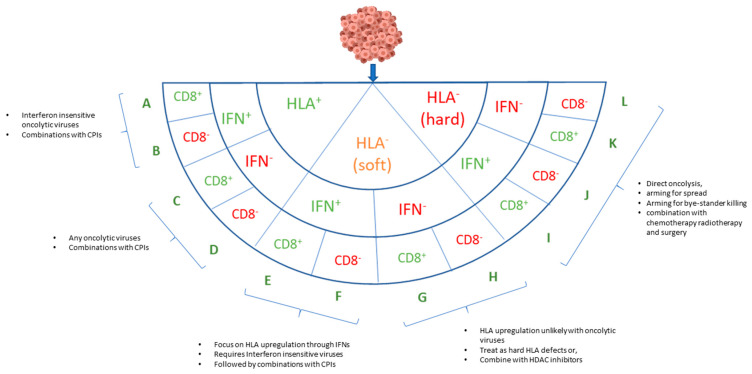
Simplified model of immunological phenotypes from the perspective of oncolytic therapy. HLA^+^ cancers (A–D) have a high potential for immune responses including oncolytic strategies to foster in situ vaccination and combinations with checkpoint inhibitors (CPIs). Cancers with a HLA^−^ (soft) phenotype (E–H) require conversion to HLA^+^ before an adaptive immune response is likely to succeed. One of the most effective ways to restore HLA expression is through interferons (IFN), which are usually expected to be an inherent byproduct of oncolysis and immunogenic cell death. Understanding the IFN status of cancer cells is of particular importance in the context of reversible HLA defects and oncolytic viruses. Cancer cells need to be IFN-competent to allow HLA upregulation, but these cells are a poor target for oncolytic viruses that are designed to selectively replicate in IFN-defective cells. With functional HLA status restored, CD8^+^ infiltration becomes an important variable. Some viruses are vulnerable to lymphocytes by design, exploiting HLA defects or immune exclusion in order to spread. Drawing lymphocytes into the tumor is an essential part of the adaptive immune response and a forte of oncolytic viruses, however lymphocytes may also eliminate some oncolytic vectors prematurely. Cancers with molecular hard defects (HLA^−^; categories I–L) or indeed soft HLA deficiencies that cannot be restored, may be difficult to treat with any therapy that relies on the adaptive immune response for efficacy. For these cancers, direct oncolysis combined with conventional cytotoxic chemoradiotherapy or the use of HLA independent killing strategies, for example encoding bi-specific T-cell engagers, may be more appropriate. Finally, it is worth noting that tumor heterogeneity means that different phenotypes may well occur within different regions of the same tumor. HDAC: histone deacetylase.

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
