# Peer review of "Tackling HLA Deficiencies Head on with Oncolytic Viruses"

_cancers, 2021, doi:10.3390/cancers13040719_

Round 1
Reviewer 1 Report
This review give a very interesting and well thought through perspective on when, where and how oncolytic virotherapy is likely to be effective, either as a standalone therapy or as an adjunct to checkpoint antibody therapy. It takes an original theoretical approach to the evaluation of oncolytic virotherapy, looking at it from the angles of HLA experession (required for adaptive antiviral and antitumor responses) and Interferon signaling (required for innate antiviral responses) by tumor cells relative to various features of the viruses used to attack the cancer. The review is written by well-respected leaders in the field of oncolytic virotherapy and will definitely be of interest to all of those working more broadly on cancer immunotherapy.
In my opinion the review is very well put together and should be acceptable for publication in its current form, but I do have a couple of suggestions for the authors to consider that might serve to further to perfect the paper and enhance its utility to the oncolytic virotherapy community.
1. It would be great to have some actual numbers indicating the percentage of human tumors belonging to the various categories that are addressed in Figure 2 (HLA+/- IFN signaling +/- and how many of the HLA+ tumors have lost HLA ABC, but retain HLA E,G,)
2. Coming from the University of Oxford, the authors should know that it is incorrect to speak of "a myriad of" something. They do this twice.
3. It would be worth stating (and possibly altering Figure 2 to reflect this) that the expression of HLA and of IFN signaling pathways is a continuum rather than a binary on/off phenomenon.
Author Response
- It would be great to have some actual numbers indicating the percentage of human tumors belonging to the various categories that are addressed in Figure 2 (HLA+/- IFN signaling +/- and how many of the HLA+ tumors have lost HLA ABC, but retain HLA E,G,)
We acknowledge that proving numbers for each of these categories would be great in principle. However, the numbers available are not all in the same indication and have not always been obtained by the same technique. Accordingly, we feel that attempting to arrange frequency data clearly in the existing figure would increase the complexity and distract from the main purpose. To address the referee’s concerns, we have emphasised that readers could refer to an excellent recent publication in this journal that covers this area in more detail (Cornel et al. 2020) (line 97) or any of references 6-10 cited in the main body of the text.
- Coming from the University of Oxford, the authors should know that it is incorrect to speak of "a myriad of" something. They do this twice.
We thank the reviewer for spotting these errors and have corrected them.
- It would be worth stating (and possibly altering Figure 2 to reflect this) that the expression of HLA and of IFN signaling pathways is a continuum rather than a binary on/off phenomenon.
Within individual cells we are not aware that this is the case, although within a population of cells within a typical tumour (and in the spectrum of tumour phenotypes) this is certainly true – accordingly we have modified the text to reflect this (lines 91-94 and 136-138).
Reviewer 2 Report
Manuscript submission # 1054679: “Tackling HLA deficiencies head on with oncolytic viruses”
This review focuses on the consequences of HLA dysregulation and loss of HLA expression in the context of oncolytic virotherapy and activation of anti-tumor immune responses, providing a thoughtful and even-handed discussion of an often overlooked and underappreciated issue. HLA dysregulation not only represents a general mechanism by which cancers escape anti-tumor immune responses, but may also impact investigational treatment strategies such as oncolytic virotherapy, which as a field has moved beyond an emphasis on the ‘brute force’ approach of maximizing viral cytolytic potency, to a more sophisticated view which involves activating the immune system to fully eradicate tumors.
A concise introduction to mechanisms of HLA dysregulation in tumors is first provided, and a “simplified model of immunological phenotypes from the perspective of oncolytic therapy” is introduced, followed by individual sections addressing different aspects of the interplay between HLA dysregulation in tumors and oncolytic viruses, including: deletion of HLA down-regulating genes from oncolytic viruses; tumor-intrinsic defects in interferon signaling and how this may impact restoring HLA expression through viral activation of interferon signaling; bypassing the loss of Class I HLA expression through the use of bispecific T cell engagers (BiTEs) and engaging non-HLA-restricted effector cells; and maximizing cell killing potency by oncolytic viruses particularly for immunologically resistant tumors whose HLA deficiency cannot be resurrected. While each individual section could itself be the subject of an entire review, the authors provide a well-balanced overview of the most pertinent aspects of each aspect.
Accordingly, overall this is a very well-written review on a topic of considerable significance, and should be considered acceptable for publication, contingent on some minor revisions to enhance clarity and completeness, and a (very) minor typo correction, as noted below:
- “The addition of checkpoint inhibitors with Imlygic26 is widely expected to further improve the response rate” (Page 3, lines 100-101): Indeed improved response rates have already been reported, as referenced (ref. #26), and currently the question is whether the combined treatment in the Imlygic–Keytruda Phase 3 clinical trial (MASTERKEY-265 / KEYNOTE-034) will demonstrate improved overall survival benefit over either alone. Also, it should be mentioned that several other clinical trials are on-going to evaluate combining immune checkpoint inhibitors with oncolytic viruses (e.g., Cavatak, Reolysin, MG1-MAGEA3, ONCOS-102, DNX-2401, HF-10, Pexa-Vec, and of course Enadenotucirev, as the authors are well aware).
- “…However, the reality is more nuanced, because defects in interferon signalling can be at different stages of the pathway, with some cancer cells not able to express interferons and others not able to respond to them. In a stromal-rich tumour, fibroblasts and macrophages are likely to have the full capacity to detect oncolytic viruses and express interferons even if the cancer cells are defective …” (Page 6, lines 217-220): These statements are certainly true enough, and accurately emphasize the complex nature of tumors and their microenvironment. However, a direct corollary of this concept, that tumors can harbor a heterogeneous population of cancer cells, is that there will likely be natural selection for cancer cells with sufficient residual interferon responsivity (even if only in response to stromal cell-derived interferon signals) to survive viral oncolysis. So the situation is not static, and while pre-treatment classification of individual cancers in different patients according to the authors’ suggested criteria may predict initial susceptibility to oncolytic virotherapy, it would perhaps be consider adding a bit more explanation that this is a dynamic situation and that tumor microheterogeneity and viral selection pressure will, perhaps likely in many cases, result in the re-emergence of resistant disease (as is indeed the case with all other monotherapies).
- “…BiTEs that retarget endogenous T cells to attack other cellular components of the tumour microenvironment…” (Page 7, lines 298-299): Depending on the target molecule against which endogenous T cells are activated, it may be prudent to include some mention of the potential need to monitor for potential “on-target / off-tumor” toxicity to normal tissues that may express the same target.
- “…5FU is a particularly attractive therapeutic for this approach because it diffuses widely and will mediate a considerable ‘bystander’ toxicity against cells that are not directly virus infected…” (Page 8, lines 356-358): It could be mentioned that another immunologically highly relevant aspect of the CD / 5FC enzyme-prodrug therapy’s bystander effect is the tumor-localized myelotoxicity caused by the 5FU generated within the tumor, resulting in killing of myeloid-derived suppressor cells (MDSC) and thereby contributing to the activation of anti-tumor immune responses (without compromising the overall immune system due to systemic myelotoxicity, as occurs with conventional chemotherapy).
- Minor correction of typo: Page 4, lines 119 (Fig. 2 legend):
“…cytoxic chemoradiotherapyor the…” --> “…cytotoxic chemotherapy or the…”
Author Response
- The addition of checkpoint inhibitors with Imlygic26is widely expected to further improve the response rate” (Page 3, lines 100-101): Indeed improved response rates have already been reported, as referenced (ref. #26), and currently the question is whether the combined treatment in the Imlygic–Keytruda Phase 3 clinical trial (MASTERKEY-265 / KEYNOTE-034) will demonstrate improved overall survival benefit over either alone. Also, it should be mentioned that several other clinical trials are on-going to evaluate combining immune checkpoint inhibitors with oncolytic viruses (e.g., Cavatak, Reolysin, MG1-MAGEA3, ONCOS-102, DNX-2401, HF-10, Pexa-Vec, and of course Enadenotucirev, as the authors are well aware).
We thank the reviewer for this insight and have change the text to acknowledge that those trials are ongoing, and also that further attention on HLA phenotypes may be helpful in interpreting outcomes. (Lines 108-117)
- “…However, the reality is more nuanced, because defects in interferon signalling can be at different stages of the pathway, with some cancer cells not able to express interferons and others not able to respond to them. In a stromal-rich tumour, fibroblasts and macrophages are likely to have the full capacity to detect oncolytic viruses and express interferons even if the cancer cells are defective …”(Page 6, lines 217-220): These statements are certainly true enough, and accurately emphasize the complex nature of tumors and their microenvironment. However, a direct corollary of this concept, that tumors can harbor a heterogeneous population of cancer cells, is that there will likely be natural selection for cancer cells with sufficient residual interferon responsivity (even if only in response to stromal cell-derived interferon signals) to survive viral oncolysis. So the situation is not static, and while pre-treatment classification of individual cancers in different patients according to the authors’ suggested criteria may predict initial susceptibility to oncolytic virotherapy, it would perhaps be consider adding a bit more explanation that this is a dynamic situation and that tumor microheterogeneity and viral selection pressure will, perhaps likely in many cases, result in the re-emergence of resistant disease (as is indeed the case with all other monotherapies).
We fully agree with this point, particularly about tumour microheterogeneity, and it is reflected in the commentary of the very first paragraph of this review (lines 25-31). We have elucidated the concept in the context of acquired resistance and also in the microheterogeneity of tumour cells accordingly (lines 88-95).
- “…BiTEs that retarget endogenous T cells to attack other cellular components of the tumour microenvironment…” (Page 7, lines 298-299): Depending on the target molecule against which endogenous T cells are activated, it may be prudent to include some mention of the potential need to monitor for potential “on-target / off-tumor” toxicity to normal tissues that may express the same target.
We agree that it may be prudent for those developing BiTEs or any other oncolytic strategy to consider the potential of on target off tumour toxicity. This concept is now discussed at lines 299-305. For this review we have tried to keep the discussion focused around HLA, although to respect the reviewer’s comments we have also softened the wording to make it less definitive about the improved toxicological profile that may be anticipated following local expression of BiTEs (line 305).
- “…5FU is a particularly attractive therapeutic for this approach because it diffuses widely and will mediate a considerable ‘bystander’ toxicity against cells that are not directly virus infected…”(Page 8, lines 356-358): It could be mentioned that another immunologically highly relevant aspect of the CD / 5FC enzyme-prodrug therapy’s bystander effect is the tumor-localized myelotoxicity caused by the 5FU generated within the tumor, resulting in killing of myeloid-derived suppressor cells (MDSC) and thereby contributing to the activation of anti-tumor immune responses (without compromising the overall immune system due to systemic myelotoxicity, as occurs with conventional chemotherapy).
We agree with this point and have amended the text to mention potential bone-marrow sparing advantages of the approach (line 371-375)
- . Minor correction of typo: Page 4, lines 119 (Fig. 2 legend):
“…cytoxic chemoradiotherapyor the…” --> “…cytotoxic chemotherapy or the…”
Thank you, we have corrected this and several other typos.
Reviewer 3 Report
In this review manuscript, the authors have reviewed studies on how to tackle HLA deficiency using oncolytic viruses. It is a well-organized review.
I have only some minor points for potential improvements.
- In a few occasions when statements were made, it seemed that authors have cited just most recent paper or just one papers, but in fact that citation of multiple (or more) papers may be more suitable. For examples:
(1). Page 3, lines 100-102. For that statement, authors have cited Ref #26. In fact, another study on the similar combination (T-VEC plus immune checkpoint blockade) for melanoma patients was published even earlier, in 2017 (Ribas A. et al., Cell 2017) (ref #41).
(2). Page 4, lines 123-125. For that statement involving ICD, the authors cited ref #29-31. In fact, one early review on ICD and oncolytic viruses was: Oncolytic Immunotherapy: Dying the Right Way is a Key to Eliciting Potent Antitumor Immunity. Front Oncol. 2014; 4:74. [PMID: 24782985].
(3). Page 4, lines 145-146. For this statement, the authors cited ref #42. Nothing wrong to cite this, but there have been quite a few previous studies along the same theme.
- Information missing in some references:
A few publishers have been publishing e-journals where articles are cited by article numbers, not page numbers. Unfortunately, their article numbers can not be downloaded automatically into Reference Management Software files. Their article numbers have to be entered manually. Theses journals include all MDPI journals, some journals from Frontiers and from Science publishers. Thus, in references #2; 5; 7; 28; 40; 69; and 71, their article numbers are missing. Please correct them by manually entering these article numbers into the Reference Management file.
- There are a few typos in the manuscript. For example, page 7, lines 262. “activity infected”. A quick document check will do the job.
Author Response
1. In a few occasions when statements were made, it seemed that authors have cited just most recent paper or just one papers, but in fact that citation of multiple (or more) papers may be more suitable. For examples:
(1). Page 3, lines 100-102. For that statement, authors have cited Ref #26. In fact, another study on the similar combination (T-VEC plus immune checkpoint blockade) for melanoma patients was published even earlier, in 2017 (Ribas A. et al., Cell 2017) (ref #41).
(2). Page 4, lines 123-125. For that statement involving ICD, the authors cited ref #29-31. In fact, one early review on ICD and oncolytic viruses was: Oncolytic Immunotherapy: Dying the Right Way is a Key to Eliciting Potent Antitumor Immunity. Front Oncol. 2014; 4:74. [PMID: 24782985].
(3). Page 4, lines 145-146. For this statement, the authors cited ref #42. Nothing wrong to cite this, but there have been quite a few previous studies along the same theme.
Thank you we have now addressed this in the revised manuscript, including all the instances mentioned above and several others. The manuscript now contains several additional references.
2. Information missing in some references:
A few publishers have been publishing e-journals where articles are cited by article numbers, not page numbers. Unfortunately, their article numbers can not be downloaded automatically into Reference Management Software files. Their article numbers have to be entered manually. Theses journals include all MDPI journals, some journals from Frontiers and from Science publishers. Thus, in references #2; 5; 7; 28; 40; 69; and 71, their article numbers are missing. Please correct them by manually entering these article numbers into the Reference Management file.
Thank you we have now addressed this and modified the references manually.
3. There are a few typos in the manuscript. For example, page 7, lines 262. “activity infected”. A quick document check will do the job.
Thank you, we have corrected this and several other typos.
Reviewer 4 Report
In the current manuscript, the authors present a review of the literature surrounding how loss of tumor cell antigen presentation might impact oncolytic virotherapy. In general, this review if seen as both highly relevant as well as quite timely. The loss of antigen presentation is likely to play a major role in many immunotherapies (including oncolytic therapy) and there does not appear to be any current reviews specifically dedicated to this subject (it is noted as a minor portion of reviews focusing on other topics). Additionally, the work is generally well laid out and well written. Despite the general enthusiasm for the overall topic, however, the specific manuscript submitted suffers from several major issues which must be addressed.
1) The major issue with the work is that it frequently fails to cite any literature supporting various points that it tries to make. As a single example, the entire section on HLA manipulating viral genes (lines 153-179) fails to cite a single paper actually describing the role that these proteins play in OV. In the absence of actually support, many of the articles claims come off more as opinion than a review of existing literature. Note that this is a fairly generalizable issue that needs to be addressed throughout the entire manuscript.
2) Similar to the point raised above, the article fails to cite any literature directly demonstrating that loss of MHC actually impacts oncolytic therapy. As a single example, has loss of B2M been shown to impact OV in any preclinical or clinical settings? If it has, this literature needs to be cited. If it has not, this should be explicitly stated as a direction that should be explored.
3) The authors correctly state that antigen presentation can be impaired through a wide variety of mechanisms. In the subsequent section (lines 180-208), however, the authors appear to focus exclusively on soft HLA deficiency caused by loss of IFN responsiveness when they suggest that OV might rescue soft antigen presentation issues. This section needs to be broadened since it is unlikely that OV will rescue many kinds of soft antigen presentation defects (for example ones derived from methylation).
4) The section on overcoming hard HLA defects is generally well written, however, it is generally overally focused on the use of BIKES/TRIKES with little attention paid to other approaches. For example, extensive work has been done examining the potential of OV to increase anti-tumor NK cells effects, however, very little of this work is cited in the current review. As this is a major path that has been explored to overcome antigen presentation defects, it should be more extensively cited.
Author Response
1) The major issue with the work is that it frequently fails to cite any literature supporting various points that it tries to make. As a single example, the entire section on HLA manipulating viral genes (lines 153-179) fails to cite a single paper actually describing the role that these proteins play in OV. In the absence of actually support, many of the articles claims come off more as opinion than a review of existing literature. Note that this is a fairly generalizable issue that needs to be addressed throughout the entire manuscript.
The reviewer makes a good point here and we have increased the number of citations where relevant and where possible. However, it should be that that there is a dearth of primary publications that specifically focus on the importance of HLA phenotype on the therapeutic outcome of oncolytic viruses and this is the main motivation behind this review. We are trying to bridge the gap by citing the wealth of HLA-research in other fields of immunotherapy, as carefully and diligently as we can, with the relative lack of information in our own field. This involves describing where those gaps lay and suggests to what might be done to address the balance. We thank the reviewer for identifying this was not as clear as it could have been and we have clarified the purpose of the review in the introduction.
2) Similar to the point raised above, the article fails to cite any literature directly demonstrating that loss of MHC actually impacts oncolytic therapy. As a single example, has loss of B2M been shown to impact OV in any preclinical or clinical settings? If it has, this literature needs to be cited. If it has not, this should be explicitly stated as a direction that should be explored.
We share the reviewer’s frustration and are not aware of any papers that focus on (or even mention) the impact of HLA phenotype on the design of oncolytic viruses or the outcome of oncolytic virus clinical trials. The definitive analysis of the most successful clinical trial (OPTIM) with Imlygic, which led to product registration, does not refer to patient HLA status at all (Andtbacka et al., Journal for the Immunotherapy of Cancer, 7, 145, 2019). This is a central motivation behind our writing the current review and we have further emphasised this point (lines 114-115).
3) The authors correctly state that antigen presentation can be impaired through a wide variety of mechanisms. In the subsequent section (lines 180-208), however, the authors appear to focus exclusively on soft HLA deficiency caused by loss of IFN responsiveness when they suggest that OV might rescue soft antigen presentation issues. This section needs to be broadened since it is unlikely that OV will rescue many kinds of soft antigen presentation defects (for example ones derived from methylation).
We agree with this comment and have emphasised that soft mutations arising from methylation of key HLA regulatory components is unlikely to be reversed by oncolytic viruses. For this reason we had mentioned possible additive activity with HDAC inhibitors, and we have now further emphasised that point in the revised text (line 78; lines 257-259).
4) The section on overcoming hard HLA defects is generally well written, however, it is generally overally focused on the use of BIKES/TRIKES with little attention paid to other approaches. For example, extensive work has been done examining the potential of OV to increase anti-tumor NK cells effects, however, very little of this work is cited in the current review. As this is a major path that has been explored to overcome antigen presentation defects, it should be more extensively cited.
We kept away from discussing natural killer cell-activating bispecific agents because – to our knowledge – despite elegant work by Miller et al. these have not yet been encoded within oncolytic viruses. In terms of the broader deployment of NK cells as therapeutic agents, the first section of the review describing approaches to overcome HLA deficiency is entirely focused on NK cells. We have also described the fact that the ability of NK cells to function in an HLA deficient environment is quite complex due to the role of HLA-G. We feel it is important to draw attention to this although there are no oncolytic virus papers that we can find that acknowledge the role of HLA-G in the context of an NK centric approach. We sincerely apologise if we have unwittingly overlooked important papers and would be happy to include them if the reviewer can point us towards them.